# From Human-Level AI Tales to AI Levelling Human Scales

## Abstract

Comparing AI models to "human level" is often misleading when model scores come from heterogeneous benchmarks or when human baselines are drawn from a narrow population. In this paper, we ask whether AI can be evaluated on a more comprehensive human-referenced scale. To address this, we propose a framework that calibrate items to the world population and report performance on a common, human-anchored scale. Concretely, we build on a set of multi-level scales for different capabilities and 'fix' the scales so that each level represents a probability of success of the whole world population on items of a given level of difficulty. As scales are defined by text rubrics with reference examples (anchors) and the base $B$, we aim at calibrating each scale for each capability (reasoning, comprehension, knowledge, volume, etc.) by compiling publicly released test items spanning education and reasoning benchmarks (PISA, TIMSS, ICAR, UKBioBank, and ReliabilityBench). The estimation of $B$ and location of anchor questions is done by extrapolating from a biased source sample (characterized by its demographics and other known information of how it was obtained) towards a larger target population (with a new demographic profile) using LLMs, with the hypothesis that they condense vast amounts of demographic data during their training. We explore different prompting mechanisms and ways to specify source and target distributions and evaluate their quality using group slicing and post-stratification on some of these datasets. The techniques introduced here allow for the definition of calibrated scales from which we can standardize other AI evaluations relative to the world population.

## 1 Introduction

Comparing artificial intelligence with human intelligence has been a constant since the early days of AI, as a way of showing progress, identifying challenges and providing intuitive information of what AI can and cannot do. However, the dominant paradigm of AI evaluation today, benchmarking, inherited from machine learning with the purpose of comparing algorithms on specific tasks, is now used to compare general-purpose AI systems such as large language models against a 'human' average (Eriksson et al., 2025; Burden et al., 2025). This collapses wide variation across skills, tasks, and populations, conflates the difficulty of the problems with the sample choice of the reference human population (often W.E.I.R.D. convenience groups), and ignores distributional structure (tails, group slices). Consequently, "human-level" claims are benchmark- and sample-dependent, and AI-human comparisons lack commensurability and granularity.

Recent evidence has shown why this matters. Reviews of human baselines in AI evaluations have documented pervasive methodological pitfalls, including small, convenience samples, inconsistent human-model test sets, and the absence of uncertainty reporting (Wei et al., 2025). In practice, this has led to contradictory headlines about AI vs human capabilities. For example, LLMs have been reported to surpass humans on certain academic benchmarks (Bojić et al., 2025) but the same models can underperform on more realistic tasks (Yeadon et al., 2024). Similarly, early theory-of-mind evaluations suggested near-human success (Kosinski, 2024), yet follow-up work revealed this apparent capability is brittle (Shapira et al., 2024) indicating reliance of LLMs on shallow heuristics rather than robust understanding. Similarly, agent-based evaluations report best systems at only 50–70% of human performance (Gou et al., 2025). In contrast, by late 2024, frontier LLMs were scoring around 90% on multi-subject academic benchmarks like MMLU (covering topics from chemistry to

law) (Phan et al., 2025) and exceeding 80% on certain graduate-level and professional exams. Many models even surpass human experts on specialized tasks: for instance, on the PhD-level GPQA Diamond science questions (Rein et al., 2024), domain experts average 70% accuracy[1], whereas state-of-the-art models now score 85–88%. Part of the contradictory results are caused by a comparison performance with humans that depends strongly on the task distribution in the benchmark, and the sample of humans used as reference. However, when items (e.g., software engineering problems) are annotated with a single scale that is normed on human populations, such as the number of hours a software engineer takes to solve the problem (Kwa et al., 2025), we can finally compare AI systems and humans more meaningfully. While this has begun to re-ground progress in human-relevant units, it does not yet provide a unified, construct-valid ruler across diverse cognitive tasks. Also, the human sample is very specific and, as any other human sample, biased.

We do not argue that humans should not be used as a reference. On the contrary, in this paper we explore more meaningful ways of doing that comparison. Actually, we suggest that the metric and the unit of measurement should not be based on performance on a benchmark but on standardised scales with human-referenced norms, for different *capabilities*. Then each value we assign to an AI system, e.g., metacognition level 2, should represent a proportion of the whole distribution of human performance for questions of that capability level, not to a single point estimate (e.g., 'average person' or 'expert') for a benchmark involving many capabilities.

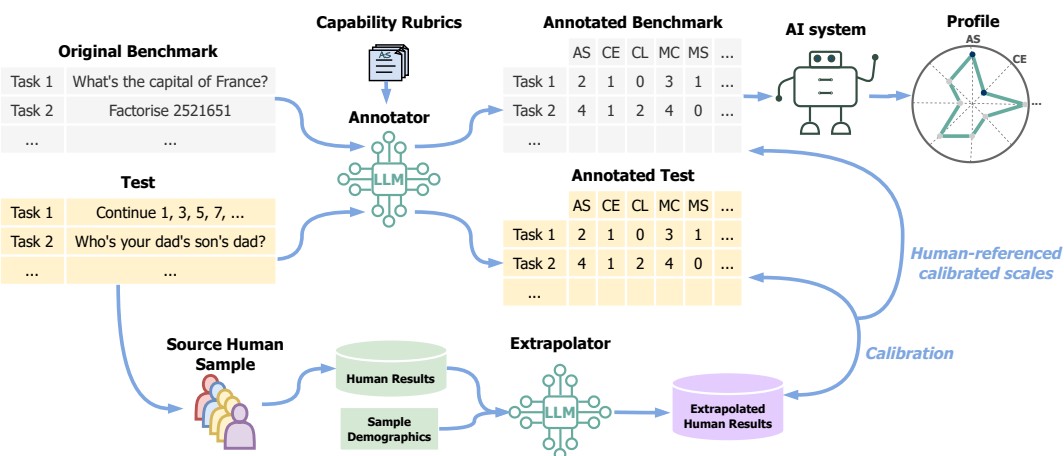

Figure 1: Calibrated annotations of benchmarks can be used to generate profiles of AI systems on human-referenced scales (top). In this paper we calibrate 18 dimensions of capability and knowledge, going from level 0 (near-universal success) to level 5 $\approx$ 1-in-$B^5$ people succeeding, with $B$ being normalized according to the human distribution taken from several tests with human results (bottom). The calibration uses this source human sample and their demographics through an LLM to extrapolate to the *whole world population* to calibrate the scales.

There are significant challenges arising from the use of human populations, some of them faced by psychology and other social sciences for a century, especially psychometrics. However, human references in AI research are more prevalent to using a sample of humans that is simply 'easy to get', such as graduate students, collaborators or merely crowdsourced data. This makes some of the criticisms about bias in collecting human population even more poignant than what has been traditionally found in psychological studies, where many studies are based on W.E.I.R.D populations (Henrich et al., 2010). Furthermore, all attempts to establish "culture-free" formats basically failed, as no measure escapes contextual bias (Gould, 1996). However, most of this criticism can be applied to the goal of characterizing a single or dominant factor of intelligence. When cognitive behavior is analyzed with a range of capability and knowledge dimensions, and samples get close to the human population, we could choose this as a norm to compare AI against, rather than a way of comparing some individuals against human groups. But still, we will always have limited and partial availability of human data, from biased samples. Inspired by ideas in equating and distribution mapping from psychometrics and sampling (Davier, 2011; Kolen & Brennan, 2013), we propose a

---

[1]https://epoch.ai/benchmarks/gpqa-diamond

lightweight approach for doing the extrapolation between a small sample and larger samples, aiming at approaching the *ideal* whole-world population (WWP), through the use of demographics and the extensive knowledge that LLMs condense about human populations.

In sum, in this paper we examine the feasibility of an ambitious vision for an AI evaluation program that can put AI capabilities on human-referenced scales, in an automated way and for all existing and future benchmarks, by the use of LLMs as annotators of capabilities and extrapolators between distributions. To this goal, we (i) use capability scales with clear construct validity, (ii) collect matched human data from partial distributions and samples, (iii) use LLMs to estimate the reference distribution (WWP), and (iv) calibrate the scales using this reference distribution. See Figure 1. This would lead to standardized scales that could be used to report commensurate, population-anchored measurements of below-human-level, human-level, and above-human-level capabilities, *for the majority of benchmarks in AI for which we do not have human data.*

To explore how feasible this vision is, we take a first step in this paper with a series of contributions:

1. We annotate the capability levels of the items of open-sourced tests from ICAR, TIMSS, PISA, UKBioBank, and ReliabilityBench with logarithmic difficulty, putting them on the same scales, as defined by the capability rubrics.

2. We introduce several mapping templates based on LLM prompts that can use the source demographics and target demographics as parameters of the extrapolation.

3. We evaluate these mappings on different slices of the human data in those tests for which we have individual demographic data, showing that for the best of them errors in the extrapolation are low, and the order of the scales is mostly preserved.

4. We use these mappings on the human data for these and the rest of tests not having demographic data, showing that we can equate different scales of difficulty via our mappings (e.g., difficulty-7 items on ICAR with difficulty-9 on TIMSS).

5. We can express meaningful and commensurate capability levels, such as frontier LLMs exceeding 90th-percentile human generalization on TIMSS math but fall below 20th-percentile reliability on PISA reading tails, highlighting non-uniform progress.

Our proposal aims to place both items and models on population-anchored capability scales that are commensurate in the sense of measurement theory. Instead of percentiles, levels increase logarithmically, taken as ratio scales as defined by Stevens (1946). Concretely, treating a wider proxy for humanity as the baseline system provides commensurate units across tasks and domains; scores become positions on a shared human reference. This makes statements like "below-human level", "human level", and "above human level" precise, supports aggregation across heterogeneous tasks, and exposes where models exceed typical humans yet remain below human tails (or vice versa). In short, by anchoring decomposed capability scales to the all-humans distribution, we convert disparate benchmark numbers into comparable measurements, and enable a principled detection of progress above and below the human baseline.

## 2 RELATED WORK

Measurement theory (Hand, 2010) builds on concepts such as units of measurement, latent variables explaining physical or social phenomena, standardized scales and commensurability. While well-established in the physical sciences, the influence on the measurement in the social sciences is also significant. Contrary to the fairy tale that measurement scales derive from ground truth, many scales and units are usually based on consensus. For instance, the metric scale for length emerged on a consensus based on a unit, the meter, initially set as a $10^{-5}$ of the distance from the North Pole to the equator, assuming an Earth flattening of 1/334, in the same way it happened with temperature, which was 'invented' as a construct (Chang, 2004). We can define scales using reference points that are familiar for humans, such as the north pole and the equator, or freezing and boiling water, and then define the operations that we can do on the scale, e.g., ensuring that differences or ratios of distances or temperature should be possible (determining an interval or a ratio scale in Stevens' taxonomy, (Stevens, 1946)). There is nothing against, in principle, taking a similar approach for cognitive capabilities and other constructs related to intelligence, which would allow us to measure natural or artificial intelligent systems on commensurate scales.

In particular, the psychometric measurement of intelligence, while taking inspiration from measurement theory, has long grappled with commensurability. Classical approaches, such as item response theory (IRT; Embretson & Reise 2000), estimate latent traits $\theta$ from observed responses but depend heavily on dense, representative data – often unavailable beyond biased norm groups. Cultural variability exacerbates this: Hofstede highlights how national cultures shape values and behaviors, affecting cognitive test validity across societies (Hofstede, 1980; Hofstede et al., 2010). Related critiques like W.E.I.R.D. sampling (Western, Educated, Industrialized, Rich, Democratic) underscore overrepresentation of atypical populations (Henrich et al., 2010), and point to how test scores often reflect agent-context interactions rather than purely latent abilities, undercutting the myth of "culture-free" testing (Ceci, 1996). Large datasets and test efforts like PISA and TIMSS, while stratified, still underrepresent global variance in education and cognitive trajectories (OECD, 2018), which can inflate or miscalibrate claims if used by AI evaluation as "human-level" capability. The ICAR suite (Condon & Revelle, 2014) offers open-source items with subgroup norms (e.g., age-/gender breakdowns), but does not provide world-population extrapolation. However, while most of psychometric approaches define scales in a populational way based on (subgroups of) humans, there is also criterion-referenced approaches Hambleton & Rogers (1991). For instance, we can determine short-term memory as how many two-digit numbers a person can remember, or use a rubric to determine different levels of complexity. However, criterion-referenced scales for different domains become incommensurate (number of numbers we can remember versus levels of metacognition). But we can perform an extra step to map them to the same reference points or units, as what Watt did with the term horsepower in a commensurate way to measure the power that a mill or a light bulb require (Hernández-Orallo, 2019).

Artificial intelligence evaluation did not evolve from human evaluation but rather from machine learning and other areas of artificial intelligence. The goal was to compare expected performance on a task for two or more competing methods, something that has driven the field for decades. But as soon as general-purpose systems like LLMs started to dominate AI, this kind of benchmark evaluation started to show problems (Eriksson et al., 2025; Burden et al., 2025). There are many other issues in AI evaluation (Hernández-Orallo, 2017; Burnell et al., 2023b; Cohn & Hernández-Orallo, 2023; Reuel-Lamparth et al., 2024), but we want to focus on the use of human baselines. Comparing against humans is critical to determine when AI can automate tasks performed by humans, or to inform about risks, especially for policy-makers and the general public, given our intuition about what humans can and cannot do. But the importance of comparing AI against human references should also remind us how many things can go wrong if human baselines are biased, simplistic or simply misconceived (Wei et al., 2025). In particular, having a human baseline on a test is not very useful when the benchmark is modified or replaced by another more difficult benchmark because of saturation (Hernandez-Orallo, 2020). In fact, it is the difficulty of the items in a benchmark which should serve to understand human baselines more properly. For instance, what is the level of logical reasoning that 30% of humans can reach? Is this level achieved by AI? One can argue that difficulty works very differently in AI and humans, and any kind of common scale that could show high correlation in the probability of success for these two different groups is wishful thinking. However, there is sufficient evidence that errors in humans and AI systems are correlated, provided we find a good proxy of difficulty. For instance, Zhou et al. (2024) probe parametric difficulty on simple tasks (e.g., addition, anagrams), showing correlation in performance and the difficulty metrics, with a very standard logistic shape. Complementary metrics, such as METR's human-anchored time-horizons (Kwa et al., 2025), quantify long-task autonomy (e.g., doubling every seven months), and also show the correlation between duration and success rate, also well modeled by a logistic function.

The use of psychometrics in AI has a long story, and IRT has long been used for analyzing populations of AI systems (Martínez-Plumed et al., 2019), in the same way as factor analysis has been used for this (Burnell et al., 2023a). However, this is usually limited to one or very few capabilities (or factors) and they derive from a population of models, a reference point that is very volatile given the pace of progress in AI. Here, we want to use a criterion-referenced approach mapped to a human reference, not a "population of LLMs" reference. We require two steps: the criterion-referenced scales and the human-norming. The ADeLe framework (Zhou et al., 2025) introduces criterion-referenced scales, by annotating items for multi-dimensional cognitive demands (e.g., quantitative-logical, attention-scan) on ratio scales (Stevens, 1946), enabling the explanation of what benchmarks measure and the demand-based prediction for anticipating model performance on new task items – yet it lacks global norming. This is what we do in this paper.

## 3 METHODOLOGY

We operationalize "human-level" as a position on population-anchored, psychometrically valid capability scales. The pipeline has five stages: (i) assemble item pools with observed human performance; (ii) annotate instance-level cognitive demands with the ADeLe methodology (Zhou et al., 2025); (iii) estimate world-population success for each item via LLM-assisted demographic adjustment; (iv) transform success rates into logarithmic, ratio-scaled difficulty levels that are commensurate across domains; and (v) validate the results by estimating total test taker population by sub-groups to gauge the exactness of extrapolation.

**Item pools and observed human performance**   We curate public, text-only items with item-level human success rates and standardized scoring. For each item, we retain the full prompt and scoring protocol, harmonize to dichotomous 0/1 scoring (collapsing partial credit when needed), and exclude items requiring visual material or with ambiguous keys or insufficient responses. Sampling-frame descriptors (age, geography, administration year, language) and subgroup identifiers are normalized across sources; when only aggregates are available, we parse per-item subgroup and full-sample rates and compute standard errors under a binomial model. These observed rates serve as ground truth for subgroup-to-sample validation and as inputs to the LLM-based extrapolation to the target world population.

**Instance-level demand annotation with ADeLe**   To preserve construct validity, we adopt the ADeLe framework (Zhou et al., 2025) to annotate each item instance along multiple capability scales. ADeLe treats demand as multi-dimensional and instance-specific: a single item can place high demand on, say, Quantitative-Logical reasoning while placing low demand on Attention-and-Scan. Concretely, we use the ADeLe v1.0 scales and public rubrics[2], which cover a broad set of 18 cognitive demands (see Table 6). Demands are assigned on a ratio scale Stevens (1946) with an absolute zero and consistent differences across levels, with rubrics designed so that doubling the demand halves the log-odds of success.

We prompt a strong LLM (i.e., GPT-5 Chat, Llama 4, GPT 4.1, DeepSeek v3.1, and GROK3) to apply the public rubrics consistently at scale, and applied to item text and scoring rules without access to model outputs. For each item $i$ and capability $c$, we obtain a vector of demand levels per item, $d_{i,c} \in \{0, 1, 2, 3, 4, 5+\}$. The result is a capability-aligned description of what the item requires, independent of who answers it.

**Instance-based calibration with LLMs**   Let $p_i^g$ be the observed success rate for item $i$ in sampling frame $g$ (e.g., a PISA wave or an ICAR cohort). Our target is to estimate the probability that a randomly drawn human from the 2025 world population answers item $i$ correctly under comparable exam conditions ($p_i^{\text{W}}$).

Estimating $p_i^{\text{W}}$ from $p_i^g$ requires an extrapolation across populations. We operationalize this as LLM-assisted post-stratification by prompting a strong language model to translate the observed rate in $g$ to the global reference W, explicitly accounting for (i) the global age distribution, (ii) education access and quality, (iii) forgetting after schooling, (iv) fluid/crystallized ability trajectories over the lifespan, (v) specialization and exposure for domain knowledge, (vi) health and cognitive decline, and (vii) language factors.

Prompts include (a) contextual introduction (situating the data set and test domain), (b) focal-group description (long-form prose containing all known demographic details $g$), (c) item content (e.g., question stems, answer choices, and correct answer), (d) observed focal-group success rate $p_i^g$, (e) reference-group description (long-form prose containing all known demographic details), (f) an explicit inference request for the model to respond with a prediction of the reference group's success rate for the current item, given the focal groups success rate and all of the demographic detail). The base prompts used for each dataset (see Table 1, Appendix A.4) appear in Table 5. Rationales are logged for auditability.

In the original ADeLe paper (Zhou et al., 2025), scales are understood with ratio-scaled difficulty using a rule of thumb of each level roughly corresponds to $L_i = \log_B(\sqrt{B}/p_i^{\text{W}})$. With a base

---

[2]https://kinds-of-intelligence-cfi.github.io/ADELE/

$B = 10$ level $L \in [0, 1)$ corresponds to 10–100% of the world population succeeding ("common" items), $L \in [1, 2)$ to 1–10% ("uncommon"), $L \in [2, 3)$ to 0.1–1% ("rare"), and so on. This provides a single, commensurate ruler across domains: equal steps in $L$ imply equal multiplicative changes in success odds in the population. However, this population is not calibrated per or across dimensions. We thus refine ADeLe's approach to pursue an LLM-prompted demographic adjustment to estimate world-population success $p_i^{\mathrm{W}}$, yielding difficulties in a logarithic scale $L_i = -\log_B(p_i^{\mathrm{W}})$.

**Validation** In the validation setting we compared extrapolations from sub-groups to the full population using four metrics, both between LLM-predicted results and real outcomes from samples with sufficient granularity in demographics. The mean absolute error (MAE) and root mean squared error (RMSE) capture the average and squared deviations between predicted and true success rates. In addition, Pearson's correlation $r$ measures linear agreement in magnitudes, while Spearman's correlation $\rho$ captures whether the relative item ranking is preserved.

## 4 EXPERIMENTS

### 4.1 DATA

We evaluate the methodology on five public sources that together cover large-scale educational assessments, open cognitive tests, a population cohort, and an LLM-oriented reliability suite. From **PISA** 2009 (OECD, 2009; Lundahl & Serder, 2020; Breakspear, 2014) and **TIMSS** 2003/2011 Grade 4/8 mathematics and science, we use released items with item-level human success rates and official sampling documentation; items requiring visual material are excluded so that all comparisons are text-only and scored 0/1 under the original rubrics. **ICAR** Letter & Number Series and Verbal Reasoning (Condon & Revelle, 2014) provide purely textual items with both participant-level responses and subgroup metadata from several cohorts, enabling robust subgroup-to-sample validation. The **UK Biobank** Fluid Intelligence test (Sudlow et al., 2015; Lyall et al., 2016; Fawns-Ritchie & Deary, 2020) contributes a timed, text-only, 13-item reasoning measure from a very large population sample. **ReliabilityBench** (Zhou et al., 2024) supplies 401 non-visual items with human success rates and human difficulty judgments across five subdomains; it lacks individual covariates and is therefore used for calibration but not subgroup validation. Table 1 summarizes these datasets, highlighting their purpose, scale, and the specific text-only subsets employed in our experiments. All items are annotated with ADeLe demand profiles using the public rubrics (Zhou et al., 2025).

Table 1: Summary of datasets and the text-only subsets used in this work.

| Dataset | Description | Scale / coverage | This work (subset) |
|---|---|---|---|
| **PISA** Lundahl & Serder (2020); Breakspear (2014) | International assessment of 15-year-olds in reading, mathematics, and science (2009); ability estimated via item response modeling with a focus on practical skills. | 57 countries; 4,500–40,000 students per country. | Text-only items: 12 math, 32 reading, 20 science. |
| **TIMSS** | Grade 4/8 mathematics & science; released items from 2003 and 2011; multiple-choice and constructed responses spanning content and cognitive domains. | 2003: G8=48 countries, G4=26 c.; 2011: G4=57 c., G8=56 c. (>360,000 students) | 300 text-only questions across 30 countries; 27 prompt variants each; evaluation subset of 5,000 variants. |
| **ICAR** (Condon & Revelle, 2014). | Public-domain cognitive tests; we use Letter & Number Series (9) and Verbal Reasoning (16), all multiple-choice and purely textual. | Validation sample 97,000 (199 countries); additional cohorts: English 145,000, Chinese 240, German 106. | 25 items used to compare item success across languages/contexts. |
| **UK Biobank** (Sudlow et al., 2015) | Fluid Intelligence (Verbal–Numerical Reasoning): 13 multiple-choice items under a 2-minute limit, probing verbal and numerical reasoning. | 502,649 participants; completed by n=168,415; 20,000 repeated after 4 years. | 13 text-only timed items as a human reference. |
| **ReliabilityBench** Zhou et al. (2024) | LLM reliability benchmark with human difficulty judgments across five subdomains; provides human success rates; all items non-visual. | 189 adults in US/UK (64% female; ages 19–78); no individual-level covariates. | 401 questions; 27 prompt variants each; used for calibration only. |

## 4.2 EXPERIMENTAL SETUP

We validate the extrapolation procedure with a unified pipeline that operates on any dataset providing item text and scoring (stems, options, keys, and scoring rules), observed item-level success rates, and metadata sufficient to define demographic subgroups.

For each dataset, we construct focal subgroups directly from the available metadata (for example, Younger/Older, Men/Women, or Country X) and compute their observed item-level success rates. The reference group is the full set of participants who attempted the item. We adopt a strict part-to-whole convention: members of a focal subgroup remain part of the reference group so that the task is to infer from a subset to the whole rather than to another subset. When a required characteristic is missing (e.g., gender not reported), those participants are excluded only from analyses that depend on that characteristic and retained elsewhere. Items with too few attempts or ambiguous scoring are filtered out to avoid unstable rates.

For every focal-group–item pair, we prompt LLMs to predict the reference-group success rate given a brief description of the dataset and test domain including the sampling frame; a prose description of the focal subgroup; the full item content and correct answer; the focal-group success rate; and a concise description of the reference group (see Sec. 3 for details). The prompt instructs the model to return a single numeric percentage for the reference group and a short rationale linking the adjustment to the item and sampling differences. To probe robustness, we paraphrase each prompt into 27 variants by reordering sections, varying connective phrasing, and altering numeric formatting, while keeping all factual content fixed.

For the LLM estimator we use a small set of instruction-tuned models spanning providers and sizes (GPT 5 Chat, GPT4.1, Llama 4, DeepSeek v3.1, and GROK3), queried with the 27 family of para-phrased prompts as described in table 5 per item with low temperature and no use of tools. For each response we parse the terminal percentage and convert to a probability. Predicted world-population rates are aggregated across item variation per model.

We validate the demographic extrapolation in a setting where the target is observable. For datasets that allow partitioning participants into demographic subgroups, we ask an LLM to infer full-sample (reference) item success rates from subgroup information within the same dataset. Subgroups are defined either from participant-level covariates (e.g., age, gender, country) or from published sub-group aggregates; in both cases, the item-wise success rates for the full sample are known and serve as ground truth. The model receives the item, the focal subgroup description and success rate, and the dataset's sampling frame, and is tasked with predicting the corresponding full-sample rate. Accurate recovery of these known totals from subgroup evidence demonstrates that the LLM performs the intended demographic adjustment, lending confidence to its use for estimating world-population success probabilities when direct observations are unavailable.

## 5 RESULTS AND ANALYSIS

### 5.1 COMPARING GROUND TRUTH/ BASELINE WITH EXTRAPOLATION

Tables 2 and 14 report validation results when models are asked to extrapolate from subgroup infor-mation to the overall population. On the ICAR benchmark, all systems achieve very low error (MAE $\approx$0.03–0.04) and extremely high correlations with the ground truth distribution (Pearson $r > 0.92$). This indicates that, when the item space is relatively homogeneous and well-structured, extrapola-tion from subgroups to the global sample is highly reliable.

Table 2: Validation results on ICAR benchmark (lower MAE/RMSE is better, higher $r$ is better).

| Model | N | MAE | RMSE | Pearson $r$ | Spearman $r$ |
|---|---|---|---|---|---|
| gpt-5-chat | 2993 | 0.030 | 0.044 | 0.976 | 0.968 |
| llama-4 | 3121 | 0.033 | 0.052 | 0.971 | 0.963 |
| gpt-4.1 | 3123 | 0.040 | 0.058 | 0.958 | 0.944 |
| deepseek-v3.1 | 3124 | 0.043 | 0.085 | 0.922 | 0.914 |
| grok-3 | 3115 | 0.043 | 0.068 | 0.939 | 0.920 |

By contrast, on TIMSS the same procedure yields substantially weaker performance: errors are three to five times larger (MAE $\approx$ 0.12–0.16) and correlations are much lower, often around 0.5–0.7. This gap suggests that the more heterogeneous content and cross-national variability in TIMSS makes extrapolation from subgroups far less accurate, and highlights important differences between benchmarks in how well subgroup-to-population generalization can be achieved.

Although current results show clear limitations, especially on heterogeneous data like TIMSS, the high correlations observed on ICAR demonstrate that models can, in principle, capture stable difficulty patterns and generalize them from subgroups to populations. This suggests that with further scale, better training data, and more explicit calibration, models may extend this capability to more complex, diverse assessments. In that sense, the present gap is not a hard limit but a challenge that can plausibly be overcome as the technology matures.

The baseline correlations between sub-samples and the full population as shown in tables 3 and 16 show that even without model extrapolation, subgroup performance aligns closely with overall outcomes. On ICAR, the agreement is exceptionally strong ($r \approx 0.95$), while for TIMSS it is notably weaker ($r \approx 0.83$), reflecting the greater heterogeneity and cross-national variability in the TIMSS data.

Table 3: ICAR ground truth / baseline — anonymized summary across focal groups.

| # Focal groups | avg N | avg MAE | avg RMSE | avg Pearson $r$ | avg Spearman $r$ |
|---|---|---|---|---|---|
| 7 | 33.143 | 0.068 | 0.086 | 0.948 | 0.894 |

## 5.2 EXTRAPOLATION OF DEMAND LEVELS

Calibration goes one step further than validation by asking models to extrapolate from the set of test takers represented in the benchmark to the distribution of the entire world population. In other words, instead of predicting overall outcomes from a subgroup, the model now estimates global capability levels from the entire available sample. Tables 4 and 15 show that this procedure is more demanding: while correlations remain reasonably strong on ICAR (Pearson $r \approx 0.93$ for the best models), errors are noticeably higher than in validation, reflecting the difficulty of scaling predictions beyond the observed sample. On TIMSS the challenge is even greater, with only one model (DeepSeek-v3.1) maintaining acceptable calibration quality, while others exhibit very high error and weak correlations.

Table 4: Calibration results on ICAR benchmark (lower MAE/RMSE is better, higher $r$ is better).

| Model | N | MAE | RMSE | Pearson $r$ | Spearman $r$ |
|---|---|---|---|---|---|
| deepseek-v3.1 | 674 | 0.166 | 0.180 | 0.933 | 0.912 |
| llama-4 | 669 | 0.168 | 0.184 | 0.920 | 0.893 |
| gpt-5-chat | 673 | 0.173 | 0.186 | 0.938 | 0.933 |
| gpt-4.1 | 672 | 0.264 | 0.282 | 0.857 | 0.829 |
| grok-3 | 671 | 0.336 | 0.355 | 0.802 | 0.780 |

These results underline that calibration is an iterative process: extrapolating overall capabilities is feasible, but current models require further refinement and methodological support to approach reliable accuracy at the population level.

Finally, we create demand levels based on Zhou et al. (2025), and change those based on the new, extrapolated outcomes, to calculate the new demand levels for world population for each of the tests[3].

---

[3]To be included in the code repository

## 6 CONCLUSION

This paper sets an ambitious program in which most benchmarking in AI evaluation could be compared against humans on a set of meaningful and commensurate capability and knowledge dimensions. For this we need criterion-referenced scales for a set of several dimensions, such as Zhou et al. (2025), fixed with a calibration of the scales to commensurate units, using the levels of the WWP on those scales. While this may look circular, it is not. We can choose the average foot size of the world-population as reference, operationalize into smaller or large units, and use that to measure the feet of people, and then the length of other objects. We explored whether this could be performed in an automated way using LLMs as our calibration tool across populations. This choice allows for a systematic, fast and reproducible way of using more sources of human data in the future to improve our calibration mappings. This also shows the potential of LLMs for this population extrapolation. Our results show that the methodology we have presented is powerful, leading to low error in the extrapolation while maintaining the ranking of human success rates, with results heavily depending on the LLM being chosen in some cases.

There are several limitations of this work. First, we have only considered a limited number of sources for human data, but it is important to highlight that openly avalaible human results on tests at the instance level and detailed demographics are unfortunately unusual. Second, we have only used a few prompting schemes with a limited number of LLMs, but this actually shows that there is margin of improvement, and the diversity of methods shows what can be done with the approaches that are time and cost-effective, compared to fine-tuning and, of course, human extrapolation or running these tests on unbiased samples of the WWP. Third, among the limitations we count the ethical issues that arise from the correct and incorrect interpretation of this work: even if our goal was to go beyond the current state of the art of biased and partial human baselines, there is always a degree of bias in our choice of datasets, extrapolation based by models that are trained on biased samples, etc. However, we show how the extrapolations can be evaluated, and this paper places the bar at a higher place that we hope new papers can criticize and improve. For more information about the ethical implications we refer to the "Ethics Statement" in the supplementary material.

This paper should allow the reunderstanding of all benchmark results in the past few years across many capabilities with a simple automated annotation procedure and without the need of any further human testing. This is not only extremely relevant for AI at the scientific and policy-making levels, but it is also relevant for the 'equating' of human populations in the social sciences.

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

# A APPENDIX

## A.1 ETHICS STATEMENT

The use of human baselines is controversial as much as it can give the impression that there is a standard or average human capability, even used as a target for the field. This misappreciation is behind blurry concepts such as human-level machine intelligence, artificial general intelligence or even superintelligence. Separating levels on several abilities allows us talk about profiles, as the diversity of people and AI systems will have very different profiles, defined by levels on these abilities.

Our paper is actually motivated by the biases and ethical issues of the use of human baselines, and in this pursuit we must also highlight some other biases that are difficult to avoid. While the whole world population is an ideal target as a reference, it can also be dominated by behavioral patterns in big countries such as India and China, not properly accounting for capability or knowledge distribution of small countries or minorities. Also, using all ages and health conditions is a moving target, and includes babies and people with age-related cognitive decline, portraying a picture of 'human-level' that may be easier to meet by AI than educated, healthy samples.

It is important to realize, though, that we use human data to fix the scales. Once this is done for all the cognitive capabilities and knowledge dimensions, we can still put groups of people in these scales, and obtain their profile. For instance, we could show that educated students from PISA results may well be above the average metacognition capability in the world population, and below in customary knowledge. Similarly, we can compare AI and different human groups on the same scale, for different dimensions. We are using the human data for fixing the scales; as the meter or the Celsius degree, the choice is not that relevant, but rather how our measurement instruments then map different individuals or groups on that scale.

## A.2 REPRODUCIBILITY STATEMENT

The data we used in this paper is already open, available and traceable through the references provided. The code and results from this paper will be available shortly after acceptance. Links to data, along with the full code and results will be available on a dedicated public github repository.

## A.3 LLM USAGE STATEMENT

LLMs were used not only as subjects of study, but as annotators and extrapolators in our methodology. This section, however, discloses the use of LLM for polishing the writing of rubrics or parts of the paper, assistance with coding, error screening and creating skeletons of reports, papers or summaries as part of this project.

## A.4 DATA

Here we include the full dataset descriptions. For each source (PISA, TIMSS, ICAR, UK Biobank, and ReliabilityBench) we document provenance and access, target populations and coverage, item formats and scoring (including IRT where applicable), inclusion/exclusion criteria (notably removal of visually dependent items), construction of prompt variants, sampling for evaluation sets, preprocessing steps, and known limitations. A high-level summary appears in Table 1.

### A.4.1 PISA

The Program for International Student Assessment (PISA) is recognized worldwide for evaluating the academic competence of 15-year-olds Lundahl & Serder (2020); Breakspear (2014). The evaluation covers reading, science, and mathematics, and assesses critical thinking, problem-solving, and the application of knowledge in these areas. It primarily focuses on practical skills rather than memorization.

The data utilized come from the 2009 survey. Its assessment encompasses 57 countries, 30 OECD members and 27 partner members (which include countries from East and Southeast Asia, Central and Eastern Europe, the Middle East, Central and South America and North Africa). Each country's test population included between 4,500 and 40,000 students. Additionally, each assessed question comes with a point score representing the estimated student's ability. This is determined using item response modeling. The relative difficulty of the questions on a test is estimated by considering the proportion of test takers who answered each question correctly OECD (2009).

For our specific experiments, we filter out items for which visual information is not necessary to answer the question. Therefore, we are not limited to multimodal models. This yields 12 math items, 32 reading items, and 20 science items.

### A.4.2 TIMSS

The *Trends in International Mathematics and Science Study* (TIMSS), conducted by the International Association for the Evaluation of Educational Achievement (IEA), is an international assessment that measures mathematics and science achievement of students in Grade 4 (approximately age 9-10) and Grade 8 (approximately age 13-14). TIMSS also collects background data (student, teacher, and school questionnaires).

The data used in this study are drawn exclusively from publicly released questions, available via the NCES "TIMSS – Released Assessment Questions" portal (see `https://nces.ed.gov/timss/released-questions.asp`). The released items include the full mathematics and science assessment items for the 2003 and 2011 waves, for both Grade 4 and Grade 8. These items cover the TIMSS content domains (for example, number, algebra, geometry, data/statistics in mathematics; life, physical, earth sciences in science) and cognitive domains (knowing, applying, reasoning). Format types among the released items include multiple-choice questions and constructed responses. These released assessment items are the basis for our analyses. In 2003, TIMSS included Grade 8 students from 48 participating countries and Grade 4 students from 26 countries, yielding a database covering more than 360,000 students worldwide. In 2011, TIMSS expanded its reach, assessing Grade 4 students in 57 countries and education systems, and Grade 8 students in 56 systems. As in previous cycles, each participating system provided nationally representative samples of students. The 2011 assessments followed the same structure as 2003, including both multiple-choice and constructed-response items in mathematics and science, balanced across content and cognitive domains.

For our experiments, as with PISA, we exclude items that rely on visual material. Since only a portion of the full TIMSS assessment is publicly released, our analyses are restricted to this subset. Nevertheless, the released items retain the structure, content balance, and difficulty profile of the complete assessment, ensuring that the resulting comparisons remain meaningful and representative. We work with a set of 300 questions distributed across 30 countries. Each question is expanded into 27 prompt variations. To construct the evaluation set, we apply stratified sampling at the country level and select a subset of 5,000 prompt variations.

### A.4.3 ICAR

The International Cognitive Ability Resource (ICAR) is constituted by a set of tests that are intended to serve as public-domain alternatives to traditional (often proprietary) human cognitive-ability measures, such as those found on commercial intelligence quotient tests (Condon & Revelle, 2014). For the present work, we focus on the ICAR's Letter and Number Series and Verbal Reasoning subtests. These tests were chosen because of their purely textual contents which could be processed without needing to deal with issues of multimodality.

The 9 items that comprise the Letter and Number Series test are similar in design to the well-known tests of the same name introduced by Thurstone (Thurstone, 1938). Test-takers must discern the patterns present in a sequence of letters or numbers and select the letter or number that completes the sequence from a set of choices. For a simple example, one might be presented with a sequence such as "$1 - 3 - 5 - 7 - ?$" and answer choices: 8, 9, 11, or 13.

The 16 items that comprise the Verbal Reasoning test include a mixture of short logic puzzles, basic arithmetic word problems, odd-one-out vocabulary items, and antonym tasks. Each item is multiple-choice with a single correct answer, designed to be solvable from the text alone without specialized background knowledge.

For the present work, the analyzed examinee responses to these tests came from two sources. The first was the original validation sample of the ICAR (Condon & Revelle, 2014). This large-scale dataset consisted of nearly 97,000 examinees (mean age 26 years, range 14–90; 66% female) from 199 countries, most of whom were students and approximately 75% of whom were located in the United States. The second data source was a multilingual validation study for an entirely different test (Gühne et al., 2021), in which participants also completed subsets of the relevant ICAR items. This latter study featured three distinct samples: a large and heterogeneous English-speaking cohort ($n \approx 145,000$, mean age 25, from over 200 countries), a Chinese secondary school sample ($n \approx 240$, mean age 15), and a smaller German university student sample ($n \approx 106$, mean age 23). Together, these data allowed for comparisons of item success rates among examinees across language and cultural contexts.

### A.4.4   UKBIOBANK

The UK Biobank is a large, population-based cohort study established to investigate the genetic, environmental, and lifestyle determinants of health in middle and older age (Sudlow et al., 2015). Between 2006 and 2010, over 500,000 adults aged 40–69 years were recruited from across the United Kingdom and completed extensive baseline assessments, including a brief computerized cognitive battery administered via touchscreen in the Assessment Centre Environment (ACE).

For the present work, we focus on the UK Biobank *Fluid Intelligence* test (also referred to as the Verbal-Numerical Reasoning test), which was selected because its items were text-based and could be processed without visual or multimodal input. This test comprises 13 multiple-choice items sampling both verbal and numerical reasoning, administered under a strict two-minute time limit, and scored as the number of correct responses (Lyall et al., 2016; Fawns-Ritchie & Deary, 2020). Items included short word problems and logical puzzles designed to probe problem-solving ability under time pressure, with no requirement for specialized knowledge.

At baseline, a large sub-sample of participants ($n = 168,415$) completed the Fluid Intelligence test (Fawns-Ritchie & Deary, 2020). The UK Biobank cohort as a whole included 502,649 participants (56% female; mean age at recruitment = 56 years, range 40–69), recruited across 22 assessment centres to capture socioeconomic, ethnic, and geographic diversity (Sudlow et al., 2015). A smaller group ($n \approx 20,000$) repeated the cognitive battery, including the Fluid Intelligence test, approximately four years later, enabling evaluation of longitudinal stability (Lyall et al., 2016). Together, these data provide one of the largest available reference samples for human performance on timed reasoning tasks.

### A.4.5   RELIABILITYBENCH

ReliabilityBench, proposed in Zhou et al. (2024), is designed to assess the reliability of LLMs. The benchmark first gathers human assessments of question difficulty across five subdomains (simple numeracy ,vocabulary reshuffle, geographical knowledge, basic and advanced science questions and information-centric transformations) and subsequently evaluates LLM performance on the same questions to establish a difficulty measure consistent with human perception. Consequently, the benchmark also records human success rates, offering human-performance data that reflect perceived question difficulty.

ReliabilityBench was taken by 189 residents in the U.S and the U.K, 64% female and 36% male, with an age range between 19 and 78 years. However, the dataset does not include individual-

level information about participants, which prevents stratification or the creation of distinct human reference groups for evaluation. As such, this dataset was only be used for calibration.

None of the questions in ReliabilityBench has visual information. Accounting for all subdomains we work with a total of 401 questions. Each question is expanded into 27 prompt variations.

## A.5 BASE-PROMPT

Table 5: Example PISA item and setup for extrapolation task.

| Section | Content |
|---------|---------|
| Introduction overall | We have PISA results from 57 countries (30 OECD + 27 partner countries). |
| Demographics | Students aged 15 years 3 months to 16 years 2 months, attending at least Grade 7 or equivalent. |
| | About 400,000–450,000 students were assessed, representing $\sim$20 million 15-year-olds globally (stratified sampling). |
| Item introduction | Consider the following question that was asked to all these students: |
| Item | *Question:* The picture shows the footprints of a man walking. The pacelength $P$ is the distance between the rear of two consecutive footprints. For men, the formula $n/P = 140$ gives an approximate relationship between $n$ and $P$, where: $n$ = number of steps per minute, and $P$ = pacelength in metres. Bernard knows his pacelength is 0.80 metres. The formula applies to Bernard's walking. **Task:** Calculate Bernard's walking speed in metres per minute and in kilometres per hour. Show your working out. |
| Percentage success rate | This question had a 19% success rate in the PISA sample. |
| Reference group | — |
| Instruction calibration | How would you translate this success rate in the PISA sample to the percentage of success that the whole world population would achieve under similar exam conditions? |
| | Mapping requires taking into account: cross-country distribution, age distribution, health conditions, access to education, forgetting, development of fluid/crystallised intelligence, acquired knowledge, cognitive decline, etc. |
| | According to these factors, and how they affect this question, what is the probability that a randomly sampled human worldwide in 2025 would answer correctly? Please give a percentage at the end. |

We iterated various versions of the, Introduction overall, Item introduction, Instruction calibration, and Instruction validation, in order to generate variation and check for robustness of results.

The full variations will be provided in the code repository.

## A.6 ADELE COGNITIVE ABILITY DIMENSIONS

Table 6 lists the ADeLe cognitive ability dimensions Zhou et al. (2025) used in our analyses, providing brief, non-exhaustive descriptions to clarify how we map task demands to abilities. For complete rubrics and scale definitions, please refer to the original ADeLe paper.

## A.7 DIFFERENTIATION TO DIFFICULTY MODELLING IN PSYCHOMETRICS

In Classical Test Theory, the "p-value", i.e., the proportion of respondents who obtain the correct response for an item, is an important index for item analysis (Rust et al., 2020). It is often referred

Table 6: Cognitive demand dimensions in the ADeLe framework. Adapted from Zhou et al. (2025).

| Dimension (Broad) | | Dimension (Specific) | | Description of Demands |
|---|---|---|---|---|
| AS | Attention and Scan | AS | Attention and Scan | Focus on or locate specific elements within a given stream of information or environment in the whole process of solving a task. |
| CE | Comprehension and Expression | CEc | Verbal Comprehension | Understand text, stories or the semantic content of other representations of ideas in different formats or modalities. |
| | | CEe | Verbal Expression | Generate and articulate ideas, stories, or semantic content in different formats or modalities. |
| CL | Conceptualisation, Learning and Abstraction | CL | Conceptualisation, Learning and Abstraction | Build new concepts, engage in inductive and analogical reasoning, map relationships between domains, and generate abstractions from concrete examples. |
| MC | Metacognition and Critical Thinking | MCr | Identifying Relevant Information | Recognise what information helps solve the task or does not, and how this recognition process unfolds as they work toward the solution. |
| | | MCt | Critical Thinking Processes | Monitor or regulate multiple thought processes to answer the question effectively, ranging from simple recall to high-level critical thinking. |
| | | MCu | Calibrating Knowns and Unknowns | Recognise the boundaries of one's knowledge and confidently identify what one knows they know, knows they don't know, or is uncertain about. |
| MS | Mind Modelling and Social Cognition | MS | Mind Modelling and Social Cognition | Model the minds of other agents or reasoning about how the beliefs, desires, intentions, and emotions of multiple other agents might interact to determine future behaviours. |
| QL | Quantitative and Logical Reasoning | QLl | Logical Reasoning | Match and apply rules, procedures, algorithms or systematic steps to premises to solve problems, derive conclusions and make decisions. |
| | | QLq | Quantitative Reasoning | Work with and reason about quantities, numbers, and numerical relationships. |
| SN | Spatial Reasoning and Navigation | SNs | Spatio-physical Reasoning | Understand spatial relationships between objects and predicting physical interactions. |
| KN | Knowledge | KNa | Knowledge of Applied Sciences | Knowledge or conceptual understanding in applied sciences (e.g., medicine, law, education, business, agriculture, engineering except IT). |
| | | KNc | Customary Everyday Knowledge | Knowledge in information that most people in a given society typically acquire through daily life experiences, social interactions, and media. |
| | | KNf | Knowledge of Formal Sciences | Knowledge or conceptual understanding in formal sciences (e.g., mathematics, logic, computer science, statistics). |
| | | KNn | Knowledge of Natural Sciences | Knowledge or conceptual understanding in natural sciences (e.g., physics, chemistry, biology, astronomy, earth sciences, ecology). |
| | | KNs | Knowledge of Social Sciences | Knowledge or conceptual understanding in social sciences and humanities (e.g., history, psychology, sociology, literature, art, philosophy). |
| AT | Atypicality | AT | Atypicality | How uncommon the task is or how unlikely it is that the instance has appeared in various sources (internet, textbooks, tests). |
| VO | Volume | VO | Volume | Proportional to the logarithm of the time a fully competent human needs to read and complete the task in ideal conditions, excluding interruptions. |
| UG | Unguessability | UG | Unguessability | The chance of error (percentage) of a task if following obvious cues or by random guess. |

to as item facility or item easiness, as it indicates the opposite of item difficulty among a particular group of respondents. According to more advanced measurement theory, for instance, Item Response Theory, item difficulty is a parameter of the item: it is defined as the point on the ability axis where the Item Characteristic Curve (i.e., a logistic curve that describes responses to a certain item given different levels of ability) is steepest. In other words, item difficulty is where a respondent needs a higher level of ability to have a good chance of getting the item correct (i.e., more than 50% probability in a 2-PL IRT model). Notably, item difficulty as defined either in Classical Test Theory or Item Response Theory is derived empirically: Item-level response data is required to calibrate

the items in order to understand how difficult each item is. This is fundamentally different from how item difficulty is operationalised in this study. Here we adopt a criterion-referenced approach, where, according to the ADeLe v1.0, 18 cognitive scales are proposed and specific rubrics corresponding to different cognitive demands are objectively defined Zhou et al. (2025). Based on these rubrics, we are able to quantify the difficulty of each item and subsequently translate it into a success rate (the p-value) in the whole world population.

## A.8 ADDITIONAL EXPERIMENTS

### A.8.1 ICAR VALIDATION AND CALIBRATION

On ICAR validation (Table 7), all models achieve very low error (MAE $\approx$0.03–0.04) and very high correlations ($r > 0.92$). Calibration (Table 8) is more challenging, with errors roughly five times larger (MAE $\approx$0.17–0.34) and correlations dropping, though the best models still maintain $r \approx 0.93$. The baseline summary across focal groups (Table 9) shows average error of 0.068 and strong correlation ($r \approx 0.95$).

Table 7: Validation results on ICAR benchmark (lower MAE/RMSE is better, higher $r$ is better).

| Model | N | MAE | RMSE | Pearson $r$ | Spearman $r$ |
|---|---|---|---|---|---|
| gpt-5-chat | 2993 | 0.030 | 0.044 | 0.976 | 0.968 |
| llama-4 | 3121 | 0.033 | 0.052 | 0.971 | 0.963 |
| gpt-4.1 | 3123 | 0.040 | 0.058 | 0.958 | 0.944 |
| deepseek-v3.1 | 3124 | 0.043 | 0.085 | 0.922 | 0.914 |
| grok-3 | 3115 | 0.043 | 0.068 | 0.939 | 0.920 |

Table 8: Calibration results on ICAR benchmark (lower MAE/RMSE is better, higher $r$ is better).

| Model | N | MAE | RMSE | Pearson $r$ | Spearman $r$ |
|---|---|---|---|---|---|
| deepseek-v3.1 | 674 | 0.166 | 0.180 | 0.933 | 0.912 |
| llama-4 | 669 | 0.168 | 0.184 | 0.920 | 0.893 |
| gpt-5-chat | 673 | 0.173 | 0.186 | 0.938 | 0.933 |
| gpt-4.1 | 672 | 0.264 | 0.282 | 0.857 | 0.829 |
| grok-3 | 671 | 0.336 | 0.355 | 0.802 | 0.780 |

Table 9: ICAR ground truth / baseline — anonymized summary across focal groups.

| # Focal groups | avg N | avg MAE | avg RMSE | avg Pearson $r$ | avg Spearman $r$ |
|---|---|---|---|---|---|
| 7 | 33.143 | 0.068 | 0.086 | 0.948 | 0.894 |

### A.8.2 PISA VALIDATION AND CALIBRATION

Validation on PISA (Table 10) reveals moderate performance: MAE in the range of 0.087–0.112 and correlations around 0.76–0.83. Calibration (Table 11) is clearly more demanding, with only DeepSeek-v3.1 retaining high correlation ($r \approx 0.94$) while other models show larger errors and reduced agreement.

Table 10: Validation results on PISA benchmark (lower MAE/RMSE is better, higher $r$ is better).

| Model | N | MAE | RMSE | Pearson $r$ | Spearman $r$ |
|---|---|---|---|---|---|
| gpt-5-chat | 18,976 | 0.087 | 0.111 | 0.825 | 0.783 |
| gpt-4.1 | 18,962 | 0.111 | 0.141 | 0.792 | 0.735 |
| deepseek-v3.1 | 18,928 | 0.111 | 0.147 | 0.763 | 0.714 |
| llama-4 | 18,946 | 0.112 | 0.147 | 0.781 | 0.730 |
| grok-3 | 18,910 | 0.112 | 0.140 | 0.818 | 0.772 |

Table 11: Calibration results on PISA benchmark (lower MAE/RMSE is better, higher $r$ is better).

| Model | N | MAE | RMSE | Pearson $r$ | Spearman $r$ |
|---|---|---|---|---|---|
| deepseek-v3.1 | 1,719 | 0.132 | 0.144 | 0.938 | 0.940 |
| llama-4 | 1,721 | 0.246 | 0.269 | 0.770 | 0.761 |
| gpt-5-chat | 1,721 | 0.257 | 0.268 | 0.902 | 0.909 |
| grok-3 | 1,720 | 0.323 | 0.340 | 0.780 | 0.784 |
| gpt-4.1 | 1,719 | 0.333 | 0.346 | 0.845 | 0.861 |

### A.8.3 UKBIOBANK CALIBRATION

For UK Biobank calibration (Table 12), performance varies strongly across models. DeepSeek-v3.1 performs best with MAE=0.184 and correlations above 0.90, while other systems show higher errors and substantially weaker alignment.

Table 12: Calibration results on UK Biobank benchmark (lower MAE/RMSE is better, higher $r$ is better).

| Model | N | MAE | RMSE | Pearson $r$ | Spearman $r$ |
|---|---|---|---|---|---|
| deepseek-v3.1 | 351 | 0.184 | 0.210 | 0.908 | 0.899 |
| llama-4 | 351 | 0.250 | 0.289 | 0.844 | 0.881 |
| gpt-5-chat | 351 | 0.265 | 0.312 | 0.781 | 0.779 |
| gpt-4.1 | 351 | 0.366 | 0.402 | 0.766 | 0.760 |
| grok-3 | 351 | 0.412 | 0.467 | 0.596 | 0.589 |

### A.8.4 RELIABILITYBENCH CALIBRATION

On ReliabilityBench calibration (Table 13), only GPT-4.1 maintains reasonable accuracy (MAE=0.103, $r \approx 0.77$–$0.85$). All other models show high error ($\approx 0.27$–$0.28$) and very low correlations, highlighting the difficulty of this benchmark.

Table 13: Calibration results on ReliabilityBench (lower MAE/RMSE is better, higher $r$ is better).

| Model | N | MAE | RMSE | Pearson $r$ | Spearman $r$ |
|---|---|---|---|---|---|
| GPT-4.1 | 10,867 | 0.103 | 0.231 | 0.771 | 0.846 |
| deepseek-v3.1 | 8,125 | 0.274 | 0.414 | 0.281 | 0.318 |
| gpt-5-chat | 8,129 | 0.278 | 0.410 | 0.228 | 0.257 |
| grok-3 | 8,131 | 0.278 | 0.417 | 0.218 | 0.257 |
| llama-4 | 8,130 | 0.279 | 0.412 | 0.245 | 0.275 |

### A.8.5 TIMSS VALIDATION AND CALIBRATION

TIMSS validation (Table 14) is substantially harder, with errors around 0.12–0.16 and correlations in the 0.5–0.7 range. Calibration (Table 15) further exposes model limitations: only DeepSeek-v3.1 retains good accuracy (MAE=0.088, $r \approx 0.87$), while other models fail to calibrate reliably.

Table 14: Validation results on TIMSS benchmark (lower MAE/RMSE is better, higher $r$ is better).

| Model | N | MAE | RMSE | Pearson $r$ | Spearman $r$ |
|---|---|---|---|---|---|
| deepseek-v3.1 | 4,689 | 0.121 | 0.149 | 0.706 | 0.663 |
| gpt-5-chat | 4,694 | 0.124 | 0.159 | 0.637 | 0.576 |
| grok-3 | 4,688 | 0.154 | 0.209 | 0.529 | 0.500 |
| gpt-4.1 | 4,698 | 0.154 | 0.205 | 0.554 | 0.511 |
| llama-4 | 4,673 | 0.164 | 0.217 | 0.550 | 0.512 |

Table 15: Calibration results on TIMSS benchmark (lower MAE/RMSE is better, higher $r$ is better).

| Model | N | MAE | RMSE | Pearson $r$ | Spearman $r$ |
|---|---|---|---|---|---|
| deepseek-v3.1 | 57 | 0.088 | 0.110 | 0.871 | 0.831 |
| gpt-5-chat | 57 | 0.432 | 0.482 | 0.435 | 0.475 |
| grok-3 | 57 | 0.435 | 0.486 | 0.073 | 0.142 |
| llama-4 | 57 | 0.445 | 0.494 | 0.441 | 0.661 |
| gpt-4.1 | 57 | 0.445 | 0.496 | 0.172 | 0.271 |

### A.8.6 TIMSS GROUND TRUTH/ BASELINE

Table 16: TIMSS ground truth / baseline - anonymized summary across all countries.

| # Countries | avg N | avg MAE | avg RMSE | avg Pearson $r$ | avg Spearman $r$ |
|---|---|---|---|---|---|
| 87 | 55.22 | 0.129 | 0.149 | 0.828 | 0.790 |

