# OpenReview forum: "From Human-Level AI Tales to AI Levelling Human Scales"
_ICLR.cc/2026/Conference — Submitted to ICLR 2026_

### Official Review · Reviewer_wK3d · 2025-10-29

**Soundness:** 1
**Presentation:** 2
**Contribution:** 2
**Rating:** 2
**Confidence:** 5

**Summary:**

This paper proposes an ambitious framework for calibrating AI evaluation on human-referenced, population-anchored scales. Building on psychometrics and measurement theory, the authors argue that “human-level AI” comparisons are unreliable due to heterogeneous benchmarks and biased human baselines. They introduce an automated methodology using LLMs to extrapolate item difficulty and success rates from limited human samples (e.g., PISA, TIMSS, ICAR, UK Biobank) to the estimated world population, yielding unified, logarithmic capability scales for comparing AI and humans. The paper demonstrates feasibility through preliminary validation, suggesting that such “human-anchored” metrics could make AI evaluation more commensurate across domains.

**Strengths:**

1. The idea of constructing population-calibrated, commensurate capability scales for AI–human comparison is novel and intellectually stimulating.

2. The paper bridges psychometrics, measurement theory, and AI evaluation, offering a new lens for understanding “human-level” claims.

3. Using LLMs for demographic extrapolation and capability annotation is an interesting and technically clever direction that may inspire further research.

**Weaknesses:**

1. The methodology is described in broad, conceptual terms, with many unverifiable steps (e.g., LLM-based demographic extrapolation). The pipeline feels speculative rather than reproducible or theoretically grounded.

2. The experiments are limited to a few datasets and report only correlation-based metrics **without strong evidence that the proposed scales produce more meaningful or reliable comparisons than existing methods**.

3. The paper reads more like a position or conceptual essay (similar to a measurement-theory manifesto). It lacks formal models, quantitative ablations, and clear algorithmic contributions expected at ICLR.

**Questions:**

None

---

> ### Author Response · Authors · 2025-12-03
> **Response To Reviewer wK3d**
>
> Q1: We will add an appendix with the exact demographic inputs provided to the LLM for WWP estimation and the full prompt templates. This was meant to be included in the released code, but will be part of the appendix as well. The rest is fully reproducible
>
> Q2: There are no baselines to compare with, this is why we do the validation in subsamples for which we have the ground truth, and then we can decide what datasets can be used for the extrapolation.
>
> Q3: We think that for ICLR it would be useful to give more background about key concepts in psychometrics and measurement theory. We acknowledge that this paper is different from the standard, incremental ICLR paper. We are attempting a challenging, new problem and we are leveraging language models for it. We found no need to use formalisations in this situation.
>
> We think the score is based on misunderstandings of the kind of paper and the objectives, asking for things (baselines) that are non-existent or non-applicable.

---

### Official Review · Reviewer_jAvT · 2025-10-30

**Soundness:** 3
**Presentation:** 3
**Contribution:** 3
**Rating:** 4
**Confidence:** 4

**Summary:**

This paper presents a framework for evaluating LLMs' capabilities on a unified, World-Wide Population (WWP)-anchored scale. It aims to move beyond comparisons to narrow human samples (e.g., specific age cohorts or WEIRD populations) or heterogeneous benchmarks. The core contribution is a five-stage pipeline combining a criterion-referenced capability framework (ADeLe) with LLM as an extrapolation tool for demographic adjustment. It takes the known success rate of a test item on a small and biased human sample as the source group, and predicts the success rate for the entire 2025 WWP. This predicted WWP success rate is then transformed into a single, commensurate difficulty scale. The authors validate the LLM's extrapolation ability on ICAR and TIMSS. They argue that such calibrated scales are standardized and hence more suitable for reporting "human-level" performance.

**Strengths:**

Originality: The paper addresses a very relevant and difficult problem in AI evaluation: the lack of a standardized, non-biased human baseline. The approach of using an SOTA LLM as a demographic extrapolation tool is quite original.

Significance: If the proposed method were proven robust, it would be significantly impactful. It introduces the concept of commensurability to AI benchmarking, allowing a reviewer to meaningfully compare a model's performance on a high-school math test (TIMSS) with its performance on a fluid intelligence measure (ICAR), all using the same human-anchored ruler.

Clarity: The paper is generally well-written and clear. The authors do an excellent job of articulating the problem of "incommensurate" benchmarks and clearly detailing the five-stage pipeline.

Quality: The paper demonstrates a good understanding of psychometrics. The experimentation is thorough, using multiple LLM models and two validation datasets.

**Weaknesses:**

1. The paper describes the ADeLe annotation process but does not provide any statistical distribution or visualization of the resulting 18-dimensional capability profiles. This makes it difficult to assess the quality or coverage of the initial scale-definition step.

2. The LLM's performance for extrapolation seems brittle. The validation on homogeneous data (ICAR) was successful, but the performance severely degrades on the more complex, heterogeneous TIMSS. Furthermore, the full "subgroup-to-WWP" calibration step failed for most models on TIMSS, indicating the method is not robust for general and cross-distribution extrapolation.

3. Using a WEIRD-trained LLM as the correction mechanism means the resulting WWP scale inherits the LLM's stereotyped model of global demographics and cognitive capability. The scale risks trading one form of human bias for another.

4. LLMs are tested in a simple, out-of-the-box fashion. There is a lack of interpretability regarding how the LLM reasons through the demographic adjustment. This makes the reported numerical results seem shallow.

**Questions:**

1. Given the failure on TIMSS, can the authors justify the use of the WWP-calibrated results derived from the other, non-validated benchmarks (PISA, UK Biobank), or should those results be treated as speculative?

2. What steps were taken to ensure high inter-rater reliability during the ADeLe annotation? Were multiple human annotators used, and what were the resulting Cohen's Kappa scores? Please also include a figure or table showing the distribution of the capability levels across the datasets.

3. Could the authors include an appendix detailing the demographic information provided to the LLM for the WWP? This is necessary to understand the "input" that supposedly guides the LLM's prediction.

---

> ### Author Response · Authors · 2025-12-03
> **Response To Reviewer jAvT**
>
> About weakness 1, this can be found in the cited paper. We deem unnecessary or even questionable to reproduce plots from another paper. In any case, it is not the capability profiles that we use here but the demand profiles.
>
> About weakness 2, performance degrades on the Trends in International Mathematics and Science Study (TIMSS. We will explain why it is so by content area, cognitive demand, question format (multiple‑choice vs. constructed response), sampling method used, and country clusters. We will explain that if the validation is not met for one dataset, this dataset should not be used. We will be more explicit about this. The important thing is that there are datasets where the extrapolation is validated, and then there is support to use it for the samples that are unknown.
>
> Regarding weakness 3, the stereotypes of the LLM will have an effect, but this is no different from extrapolations made by humans and the access of the characteristics that we have. The validation shows this can work for some datasets, and those are the ones that should be used for the extrapolation.
>
> Q1: Interpreting heterogeneous results. We will justify which datasets support confident calibration (e.g., more homogeneous test domains) and which should be treated as preliminary, using the existing ICAR vs. TIMSS contrast to motivate this distinction.
>
> Q2: Same as weakness 1, these results are in the cited paper. We will be explicit about where to find them and we will give a summary (and that they are really high), but there’s no need to copy the data or plots from another paper.
>
> Q3: Transparency on inputs. We will add an appendix with the exact demographic inputs provided to the LLM for WWP estimation and the full prompt templates. This was meant to be included in the released code, but will be part of the appendix as well.

---

### Official Review · Reviewer_sJxH · 2025-10-30

**Soundness:** 2
**Presentation:** 1
**Contribution:** 1
**Rating:** 2
**Confidence:** 4

**Summary:**

The submission proposes a methodology that attempts to obtain a multifaceted measure of AI versus human performance using large-scale datasets derived from human performance on certain standardized scholastic and psychometric tests. The goal of the paper is to obtain a standardized scale for each measure of intelligence that is derived from the fraction of humans that would solve a given task correctly. Since any given task may require different aspects of intelligence, it is also necessary to annotate every task with the demand it places on each aspect intelligence. This annotation is based on publicly available rubrics designed in prior work on measuring multifaceted aspects of intelligence. Finally, for any given dataset based on human performance on scholastic or psychometric tests, there is usually significant demographic bias. Hence, the submission proposes a method to extrapolate from demographically biased samples to the true demographic distribution of the whole world population.

Both the annotation of tasks with the demands on intelligence and the extrapolation to the world population are done with LLMs. For annotation the LLMs are prompted to follow the publicly available rubrics. For extrapolation, LLMs are directly prompted to predict the world population performance on a task given the sample performance along with demographic information about the world population and the sample population. Experiments with subsamples of large datasets are used to measure how well these LLM assisted annotation and extrapolation steps work empirically.

**Strengths:**

Building large-scale human-normalized measurements of LLM cognitive performance is an important question, and the idea of using LLMs to assist in key aspects of building such a novel measurement is interesting.

**Weaknesses:**

1. The empirical results seem quite mixed, LLM demographic extrapolation did not seem to perform consistently well across different datasets.
2. The empirical setup lacks adequate baselines. Extrapolating from a biased sample population to a larger population with known demographics is a very well-studied problem in statistics. For instance, all effective political polling uses statistical methods to make such adjustments based on predicted voter turn-out rates. The use of a LLMs to make such predictions based on demographic information needs, at a minimum, to be compared to the standard techniques from statistics that are designed to accomplish the same task.
3. The writing and organization of the paper are a significant impediment to understanding. There are many convoluted sentences and undefined terms that the reader is forced to struggle with in the beginning of the paper. The related work section is also quite strange, with an entire paragraph about how measurements of physical quantities arose in much earlier historical context, but no mention of the (highly relevant) standard statistical methods used to extrapolate from a biased sample based on demographics.

**Questions:**

How would standard methods like importance weighting based on demographic data perform on the extrapolation task?
Why would one expect LLMs to outperform rigorous statistical methods for what is fundamentally a problem of basic statistics?

---

> ### Author Response · Authors · 2025-12-03
> **Response To Reviewer sJxH**
>
> There is a misunderstanding in what we extrapolate, compared to the validation case. In the validation case we have the proportions of hold-out populations, so we could use techniques such as iterative proportional fitting [IPF]), inverse probability weighting (IPW), and multilevel regression with post‑stratification (MRP) where covariates and sample sizes permit. But then we could not extrapolate for the WWP, as we do not have a full characterisation in the same parameters as each dataset. As each dataset has a different characterisation, this would not only be inapplicable but infeasible. So it is not the standard extrapolation from a biased sample where we have non-representative proportions of each subgroup. We will clarify this.
>
> If the score is based on this misunderstanding, we believe it is unfair. If the score is based on organization, we can restructure the paper to define the estimator clearly, list assumptions and failure modes, move background material to where it supports the argument, and improve figure clarity and captioning.

---

### Official Review · Reviewer_oG5r · 2025-11-02

**Soundness:** 2
**Presentation:** 3
**Contribution:** 2
**Rating:** 4
**Confidence:** 4

**Summary:**

To address misleading comparisons between AI and narrow human benchmarks, this paper introduces a framework to calibrate AI performance against the entire world population. The core contribution is a novel technique that uses Large Language Models (LLMs) to extrapolate data from existing, biased source samples (like PISA or UKBioBank) to this broader, global scale. This methodology allows for the definition of "calibrated scales," enabling AI evaluations to be meaningfully standardized relative to the global population rather than unrepresentative test groups.

**Strengths:**

1.The paper tackles a recognized and significant flaw in AI evaluation: current comparisons to "human level" are often misleading due to heterogeneous benchmarks and narrow, unrepresentative human baselines (e.g., W.E.I.R.D. populations). The motivation to correct this is well-justified and highly relevant.
2.The paper proposes an ambitious framework to move beyond fixing local benchmarks, aiming instead to calibrate AI evaluation onto a single, commensurate scale anchored to the "world population". The core novel method is the use of LLMs as "demographic extrapolators," leveraging their condensed knowledge to map biased source samples to a global distribution.
3.The study provides empirical validation of its approach. The results on ICAR dataset are particularly strong, showing that the LLM extrapolation from subgroups to the full sample achieves very low MAE (0.03-0.04) and extremely high correlation (Pearson r > 0.92).

**Weaknesses:**

1.You have pointed out the core methodological gap in the paper. The experiment's "validation" setting only proves that the method can extrapolate from a biased subgroup to that dataset's full population (e.g., from "males aged 20-30 in ICAR" to "all participants in ICAR"). However, that "full population" from the dataset is still a biased sample and is not representative of the "whole world population" (WWP).Therefore, the experiment cannot validate the final, most important step: the "calibration," which attempts to extrapolate from that biased dataset to the ideal WWP. This final step remains an unverifiable assumption because no ground truth exists for the WWP to measure against.
2. A significant weakness of this methodology is its fundamental reliance on the Large Language Model's intrinsic capability to perform demographic extrapolation . This introduces a critical and potentially circular flaw: the LLMs themselves are trained on datasets that are notoriously biased and in no way representative of the "world population." The vast majority of their training data is sourced from the internet and is heavily skewed toward English-language content and WEIRD (Western, Educated, Industrialized, Rich, and Democratic) populations. Therefore, when the method prompts the LLM to estimate the success rate for the "whole world population" by accounting for global factors , it is not drawing from a neutral, ground-truth understanding of humanity. Instead, it is highly likely that the LLM is projecting its own inherent training biases onto the extrapolation. This "bias-in, bias-out" problem undermines the core claim, as the method risks simply replacing the known bias of the source sample with the unknown, but significant, bias of the LLM itself.

**Questions:**

1.Could you provide a more in-depth diagnostic analysis explaining why the model performs poorly when extrapolating TIMSS data? Is this failure due to content heterogeneity, cross-cultural differences, or a lack of specific data in the LLM knowledge base? More importantly, if the method fails when dealing with complex, heterogeneous data, does this fundamentally undermine its core value as a general calibration tool to address current benchmark heterogeneity issues?
2.Can the authors provide additional theoretical or empirical evidence (e.g., using large datasets as simulations of “proxy worlds”) to demonstrate why we should believe that success on the “verification” task (as shown by ICAR) can translate into accuracy on the final, completely unverifiable “calibration” task?
3.When prompted to consider factors such as global age distribution, education level, and health status, how can we be sure that LLM is not "guessing" or "adjusting based on stereotypes," thereby introducing a new, systematic bias? Can you demonstrate that the results "calibrated" by LLM are closer to the true "global population" than the original biased sample, and not just "another form of bias"?

---

> ### Author Response · Authors · 2025-12-03
> **Response To Reviewer oG5r**
>
> We fully recognize there’s no available ground truth about the whole world population. This is exactly the gap we want to close with this paper. In order to understand this, we will better clarify the distinction between validation of our method (verification with the known truth) and the extrapolation to the unknown (and its use for calibration). We will make explicit that our current “validation” step (from sub‑group to full sample within a dataset) already serves as a proxy‑world check, and we will present item‑level and slice‑level details rather than only aggregates. The hypothesis is that if the methodology, with the inherent bias in the LLMs that are used for extrapolation, is good for extrapolating inside a dataset (where we have some ground truth), then we expect a meaningful extrapolation to the final world‑wide population (WWP). The larger the number of datasets and the more diverse they are in evaluation and demographics, the more it allows us to determine the biases that come from the original datasets, based on the LLM-based extrapolation method we used. It is precisely because there is no ground truth about the whole world population, why we need this. In the same way, we can make estimates of many other quantities and distributions in other sciences: from the number of mosquitoes in a country or the number of stars in the galaxy, which are unknown but can be estimated. We derive methods and apply them in subsets of the space where we do have ground truth, validate them with it, and then we use them to extrapolate to the areas where we don’t have ground truth. This is how experimental sciences work.
>
> The same argument about circularity could be applied to any experimental science, as we only capture information from part of the galaxy and then our theories are biased to what we can observe, or even “circular”. In any case, the final WWP estimation and the associated calibration will always come with an estimate of uncertainty (isomorphically, any measurement comes with measurement errors, and any prediction/ model with prediction/ estimation errors).
>
> TIMSS diagnostics. We will analyze where performance degrades on the Trends in International Mathematics and Science Study (TIMSS): by our sampling method, content area, cognitive demand, question format (e.g., multiple‑choice vs. open response), and country clusters. This will explain why the method is brittle on this dataset in particular. We will explain that if the validation is not met for one dataset, this dataset should not be used. We will be more explicit about this. The important thing is that there are datasets where the extrapolation is validated, and then there is support to use it for the samples that are unknown.
>
> The reviewer is asking us for something impossible. There is no way of providing theoretical or empirical evidence that the WWP estimation is correct. What we do is to validate that the extrapolation works in subpopulations, even if the LLM may be "guessing" or "adjusting based on stereotypes". We will show, that by asking the models about their reasonings, this even opens opportunities for evaluation of said stereotypes and improvement of fairness in future models.
>
> Overall, we will be more clear about the WWP values to be estimates that will be done by different methods, and changes as we obtain more evidence. This is the first method that tries to do this based on empirical models. The only alternative way is asking human experts to do this, but the biases are going to be similar or larger than with LLMs, as they have an even more limited access to demographic data (and may e.g., involuntarily argue based on traditioned yet outdated theories about inequality of humans, which originate from colonial times - see multiple retractions for country based IQ).

---

### Author Response · Authors · 2025-12-03
**General response to all reviewers**

We thank the reviewers for their reviews but many of them are misinterpreting the paper or requesting things that do not make sense in this context. Our aim is to place artificial‑intelligence results on human‑anchored, commensurate scales using open human data; this submission is an early step of a larger research endeavor, with real results. We have clarified these misunderstandings in the responses to the reviews but we will modify the paper to avoid these misunderstandings by completing the information or clarifying the problem and the solution. Concretely, we will (i) more clearly separate what we do in validation (where we have ground truth) and, only when the results are positive, we can take the extrapolation to the world‑wide population (WWP) values as reasonable. Still, we will provide estimates with uncertainty ranges. We will do more partial checks (including item‑ and slice‑level details rather than only aggregates); (ii) explain why standard survey‑sampling/statistical baselines for subgroup‑to‑population adjustment are not feasible; (iii) provide targeted diagnostics for heterogeneous assessments - especially Trends in International Mathematics and Science Study (TIMSS) - by country clusters, if possible, content area cognitive demand, question format; (iv) tighten writing and organization with a compact estimator section (inputs, outputs, assumptions, limitations) and clearer figures. These steps respond directly to the recurring concerns and are motivated by the present pattern in the results (strong verification on International Cognitive Ability Resource [ICAR], brittleness on TIMSS).

---

### Meta-Review · Area_Chair_YgJL · 2026-01-07

**Summary:**

This paper presents an automated methodology that uses LLMs to extrapolate item difficulty and success rates from limited human samples (e.g., PISA, TIMSS, ICAR, and the UK Biobank) to an estimated world population, yielding unified logarithmic capability scales for comparing AI systems and humans.
The idea and motivation are quite interesting; however, the paper is not well organized or well presented for an ICLR submission. The methodology is described in broad, largely conceptual terms and requires more rigorous experimental validation to demonstrate the advantages of the proposed approach.

**Reviewer Concerns:**

The methodology description is largely conceptual and informal, and the experimental comparisons are limited. The reported numerical results appear to lack depth, and this work would benefit from clearer organization and presentation. During the rebuttal, the authors provided some clarifications; however, the additional evidence was not sufficiently persuasive.

**Reviewer Scores:**

The rebuttal clarifications may warrant a slight score increase, but they are not sufficient to change the acceptance decision.

---

### Decision · Program_Chairs · 2026-01-26

Reject